# Intake of MPRO3 over 4 Weeks Reduces Glucose Levels and Improves Gastrointestinal Health and Metabolism

**DOI:** 10.3390/microorganisms10010088

**Published:** 2021-12-31

**Authors:** Songhee Lee, Heesang You, Yeongju Lee, Haingwoon Baik, Jeankyung Paik, Hayera Lee, Soodong Park, Jaejung Shim, Junglyoul Lee, Sunghee Hyun

**Affiliations:** 1Department of Biomedical Laboratory Science, Graduate School, Eulji University, 712, Dongil-ro, Uijeongbu-si 11759, Korea; song-1107@naver.com (S.L.); dlddwn1@gmail.com (Y.L.); 2Department of Senior Healthcare, Graduate School, Eulji University, 712, Dongil-ro, Uijeongbu-si 11759, Korea; yhs1532@nate.com; 3Department of Biochemistry and Molecular Biology, Graduate School, Eulji University School of Medicine, Daejeon 34824, Korea; baikhw@eulji.ac.kr; 4Department of Food and Nutrition, Graduate School, Eulji University, Seongnam 13135, Korea; jkpaik@eulji.ac.kr; 5R&BD Center, hy Co., Ltd., 22, Giheungdanji-ro 24beon-gil, Giheung-gu, Yongin-si 17086, Korea; yera@hy.co.kr (H.L.); soodpark@hy.co.kr (S.P.); jjshim@hy.co.kr (J.S.); jlleesk@hy.co.kr (J.L.)

**Keywords:** gut microbiota, 16S rRNA, probiotics, prebiotics, synbiotics

## Abstract

Human gut microbiota are involved in different metabolic processes, such as digestion and nutrient synthesis, among others. For the elderly, supplements are a major means of maintaining health and improving intestinal homeostasis. In this study, 51 elderly women were administered MPRO3 (*n* = 17), a placebo (*n* = 16), or both (MPRO3: 1 week, placebo: 3 weeks; *n* = 18) for 4 weeks. The fecal microbiota were analyzed by sequencing the 16S rRNA gene V3–V4 super-variable region. The dietary fiber intake increased, and glucose levels decreased with 4-week MPRO3 intake. Reflux, indigestion, and diarrhea syndromes gradually improved with MPRO3 intake, whereas constipation was maintained. The stool shape also improved. *Bifidobacterium animalis*, *B. pseudolongum, Lactobacillus plantarum*, and *L. paracasei* were relatively more abundant after 4 weeks of MPRO3 intake than in those subjects after a 1-week intake. *Bifidobacterium* and *B. longum* abundances increased after 1 week of MPRO3 intake but decreased when the intake was discontinued. Among different modules and pathways, all 10 modules analyzed showed a relatively high association with 4-week MPRO3 intake. The mineral absorption pathway and cortisol biosynthesis and secretion pathways correlated with the *B. animalis* and *B. pseudolongum* abundances at 4 weeks. Therefore, 4-week MPRO3 intake decreased the fasting blood glucose level and improved intestinal health and metabolism.

## 1. Introduction

The human gut microbiota is a complex ecosystem comprising numerous bacteria that colonize the gut [1]. Of more than 10 million microorganisms discovered to date in humans, 1500 were found to be bacterial species prevalent in the human gut [2]. These bacterial species are also collectively referred to as the “gut microbiome” because it contains a vast collection of genes and is currently recognized as a secondary genome [3,4]. The gut microbiota is involved in different human metabolic processes, such as absorption, digestion, nutrient synthesis, and metabolism of the immune system [5]. It has also been implicated in the pathogenesis of various metabolic diseases, including diabetes, obesity, and inflammatory bowel disease, and is considered to be widely involved in human physiology and metabolism [6,7,8].

Compared to younger adults, older adults have a lower glycolytic metabolic capacity and increased putrefactive metabolic activity [9]. With aging, there is a failure of homeostasis in functions such as nutrient absorption, gastrointestinal (GI) tract motility, and insulin secretion, and the efficiency of these functions deteriorates [10,11]. Representative features of the gut microbiota of the elderly include a decrease in the overall diversity of bacteria and the abundance of beneficial microorganisms [12]. Changes in the gut microbiome are not synonymous with detrimental changes in health, but with respect to aging, these changes can indicate worsening health [13]. Conversely, the moderate intake of dietary fiber and increase in fermentation by beneficial bacteria in the large intestine can be considered helpful. These not only increase intestinal motility but also contribute to the overall well-being of individuals by strengthening the intestinal barrier, which, in turn, helps attenuate pro-inflammatory conditions. The composition of the gut microbiota is influenced by dietary habits and composition. Diet and lifestyle may be the most important contributors to changes in bacterial composition in people of all ages, especially the elderly [12,13].

According to the World Health Organization, probiotics are defined as “living microorganisms that, when administered in appropriate amounts, confer a health benefit on the host” [14]. Probiotics can inhibit gut pathogens, interact with the gut microbiome, and modulate the immune system, either directly or through modification of the gut microbiota. The most commonly used probiotics are *Bifidobacterium* and *Lactobacillus* [15]. The intake of probiotics was shown to reduce the severity of metabolic syndrome in animal experiments, and positive effects on body weight, glucose levels, and insulin metabolism were predicted in clinical trials [16]. Prebiotics are defined as “selectively fermented ingredients that produce specific changes in the composition of the GI microbiota that provide benefits to host health” [17]. Most prebiotics are found in whole grains, fruits, and vegetables. Complex carbohydrate-like compounds present in the small intestine or produced commercially in probiotics are transported to the large intestine without being digested or absorbed, where selective fermentation promotes changes in the composition and metabolic activity of the gut microbiome [18]. The use of probiotics and prebiotics supplements is a major means of maintaining health and a nutritional strategy for improving intestinal homeostasis [19].

However, despite the growing interest in the food industry among the elderly, there is a lack of intervention studies demonstrating the effectiveness of specific probiotic substrates in the elderly and explaining the causal relationship between probiotic substrate intake and changes in the microbial community composition [20]. Among the many studies conducted on commercial strains of the genera *Lactobacillus* and *Bifidobacterium* over the past decade, only a few have tested the effects of probiotics on the gut microbiome. Although there has been a focus on evaluating the efficacy of these strains on GI disorders associated with antibiotic consumption and disease [20], there remains the need for standardized, reproducible, and comparable intervention protocols for investigating the efficacy of probiotics and prebiotics in the elderly.

The MPRO3 used in this study is a form of synbiotics; that is, a mixture of probiotics and prebiotics. Probiotics are composed of *Bifidobacterium animalis* spp., *Lactobacillus casei*, and *Lactobacillus plantarum*, and prebiotics include dietary fiber (polydextrose, chicory dietary fiber, lactulose, and wheat dietary fiber) and functional oligosaccharides (fructo-oligosaccharide, isomalto-oligosaccharide, and of xylo-oligosaccharides). We studied the effect of probiotic intake for three weeks in a previous study [19]. However, no studies have been conducted on its effectiveness over periods shorter or longer than three weeks. In this study, at the baseline time point (zero weeks), the group was divided into two groups: a group that would take MPRO3 (synbiotics) and a group that would take a placebo. The placebo group took the placebo for the total duration of the study, four weeks. After taking MPRO3 for one week, the MPRO3 group was further divided into one group that continued MPRO3 intake for three weeks and another group that stopped taking MPRO3 and continued with the placebo for three weeks. Our first purpose was to observe whether the intake of synbiotics for a short period of one week can induce changes in the intestinal microbiota. In addition, by observing whether the microbiota returns to the same state as before after one week of taking synbiotics, we intend to prove the need for a steady intake of synbiotics for the elderly.

This study aimed to evaluate changes in blood parameters, defecation activity, and intestinal microbiota composition in the elderly cohort who received synbiotics (a combination of probiotics and prebiotics). In addition, this study aimed to observe and compare the effects of synbiotics in terms of physiological activity after one and four weeks of intake, as well as after no intake of synbiotics.

## 2. Materials and Methods

### 2.1. Participants

Between April and June 2021, 54 participants were recruited for the study. The participants were recruited based on the following inclusion criteria: (1) females, (2) age greater than or equal to 65 years, (3) willingness to participate in the study, obtain consent if necessary, voluntarily consume the study subject, and sign the informed consent form. The exclusion criteria were as follows: (1) use of antibiotics, (2) diagnosis of a mental illness, (3) use of antipsychotics, and (4) diagnosis of GI disorders and the use of medications, including antibiotics. All participants provided consent after understanding the study purpose, examination schedule, and potential risks. This study was conducted in conformance with the ethical guidelines of the Declaration of Helsinki. The study was approved by the Eulji University Internal Review Board (IRB No. EUIRB 2021-008).

### 2.2. Intervention Study Design

The study had a randomized single-blind design, using a blocked randomization method [21], and was conducted over a time span of 4 weeks. Prior to the commencement of the study, participants were restricted from consuming dairy products for 3 weeks, for irrigation of the GI tract. The participants were randomly divided into MPRO3 intake Groups A and B and the placebo Group C, based on the intake of different substances during the first week of the 4-week intake period. During the 3 subsequent weeks, the participants consumed MPRO3 (Group A), discontinued MPRO3 consumption and instead consumed the placebo (Group B), or only consumed the placebo (Group C) (Figure 1). The measurements and tests were performed identically for a total of three visits. We used a survey questionnaire based on 24-h dietary recall evaluation, the gastrointestinal symptom rating scale (GSRS), and the Bristol stool form scale (BSFS). For each participant, a physical examination was performed, blood pressure was measured, and a blood test was performed for evaluating the hematological and chemical parameters. We analyzed the zonulin levels and next-generation sequencing profiles in the stool samples.

### 2.3. Ingredients and Intake of MPRO3

The synbiotic drink used in this study was a commercial product (MPRO3, hy Co. Ltd., Yongin-si, Gyeonggi-do, Korea), with one bottle containing two capsules (130 mg per capsules) and 130 mL of a solution. The capsule and solution were in a ratio of 1:2 and contained 10 billion colony-forming units (CFUs) of bacteria. The bacterial count was 5.0 × 10^9^ CFUs of *Bifidobacterium animalis* spp. lactis HY8002, 2.5 × 10^9^ CFUs of *Lactobacillus casei* HY2782, and 2.5 × 10^9^ CFUs of *Lactobacillus plantarum* HY7712. The solution (130 mL) contained dietary fiber (polydextrose, chicory dietary fiber, lactulose, and wheat dietary fiber) and functional oligosaccharides (fructo-oligosaccharides, isomalto-oligosaccharides, and of xylo-oligosaccharides). The synbiotic drink and placebo had the same appearance and taste. The other components are listed in Appendix A.

Participants consumed the drink from one bottle before meals, on an empty stomach, each morning for 4 weeks. They were instructed to consume the drink from one bottle per day at the same time of day. Adherence to intake was monitored daily by the investigator on a one-on-one basis via telephone. To maintain the freshness of MPRO3, evaluate the compliance of the participants with the study methods, and prevent confusion among multiple groups, the product was provided each week, and not all at once, for 4 weeks. Participants were advised to consume only the MPRO3 dairy product provided during the study period and maintain their usual diet.

### 2.4. Survey Questionnaire

Dietary intake was assessed using the 24-h recall method, and nutrient intake was analyzed using the computer-aided nutritional analysis program for professionals (CAN-Pro 5.0 program; Korean Nutrition Society, Seoul, Korea) [22]. We also administered a questionnaire on family history, medical history, diet, smoking, drinking, antibiotic use, current medications, nutritional supplement intake, and adverse reactions (Appendix A).

The GSRS is a disease-specific instrument that includes 15 items categorized into five symptom clusters, addressing different GI symptoms. The five symptom clusters include reflux, abdominal pain, indigestion, diarrhea, and constipation. The GSRS uses a seven-grade Likert-type scale, in which 1 represents the absence of bothersome symptoms and 7 represents highly bothersome symptoms: a score of 1 represents “no problems”, and a score of 7 represents “severe problems”. The severity of symptoms may be graded as no problems (1 point), mild (1–2 points), moderate (2–4 points), or severe (4–7 points) [23].

The BSFS is one of the most widely used scales in both clinical and research settings. The BSFS is a Likert scale used to classify stool forms into seven categories, and it has been validated as a surrogate measure for GI transit time as a diagnostic tool designed to classify the shape and type of human feces into seven distinct categories: separate hard lumps, like nuts (=1); sausage-shaped, but lumpy (=2); like a sausage, but with cracks on its surface (=3), like a sausage or snake, smooth and soft (=4); soft blobs with clear-cut edges (=5); fluffy pieces with ragged edges, mushy stool (=6); and watery, no solid pieces, entirely liquid (=7) [24].

### 2.5. Anthropometric Measurements

During weight measurement, the participants removed their shoes and heavy fabrics. During height measurement, the participants aligned their shoulder blades, hips, and heels, leaned against a wall, and maintained their necks in a natural, unextended position. During body circumference measurement, the circumferences of the middle part of the upper arm, calf, waist, and hip were measured using a non-stretchable flexible measuring tape. Participants stood with their feet joined and arms on their sides. Hip circumference was measured at the maximum hip circumference. The body mass index was estimated by dividing the weight (kg) by height squared (m^2^). The waist-to-hip ratio (WHR) was estimated by dividing the waist circumference by the hip circumference. The threshold WHR was ≥0.85 for women and ≥1.00 for men, and a WHR greater than these values were considered to indicate excellent adipose tissue distribution [25].

### 2.6. Blood Tests

#### 2.6.1. Hematological and Chemical Tests

Blood samples were collected from the antecubital vein of the arm using a vacuum tube [26]. After collection, the blood was inverted two to three times and mixed. Plasma and red blood cells were separated from the sample via centrifugation (1500× *g*, 4 °C, 15 min). EDTA Vacutainer blood collection tubes for hematological tests and SST Vacutainer blood collection tubes for biochemical tests (Becton Dickinson, Franklin Lakes, NJ, USA) were transferred to a hospital (Central Hospital, Seongnam-si, Korea). The samples were immediately analyzed using standard laboratory methods and certified assays. Residual blood was stored at −80 °C.

#### 2.6.2. Zonulin Measurement

To measure the expression of zonulin in blood serum, we used the human zonulin ELISA Kit (Cusabio, Wuhan, China) according to the manufacturer’s instructions [21]. Briefly, 100 μL of the samples and standards from the kit were added to a 96-well microplate pre-coated with an antibody specific for zonulin and incubated for 2 h at 37 °C. Next, 100 μL of biotin-antibody (1×) was added and incubated for 1 h, followed by the addition of 100 μL of horseradish peroxidase–avidin. After 1 h, 90 μL of 3,3′,5,5′-tetramethylbenzidine substrate was added and incubated for 30 min at 37 °C. Finally, 50 μL of stop solution was added, and the absorbance at 450 and 570 nm was measured using a microplate reader.

### 2.7. Feces Profiling

#### 2.7.1. Feces Testing and DNA Extraction

Fecal samples collected on the day of the visit were transferred within 2 h to a sealed shipping box that was maintained at −4 °C, which was followed by analysis within 4 h in the laboratory and storage of the samples at −80 °C until further analysis. Two hundred and fifty milligrams of the stool samples were used for DNA extraction. DNA extraction was performed using the QIAamp PowerFecal pro-DNA kit (Qiagen, Hilden, Germany) according to the manufacturer’s recommendations [27].

#### 2.7.2. Bacterial 16S rRNA Polymerase Chain Reaction (PCR) Amplicon Library Preparation

Sequencing was performed using the Ion Library fabrication kit on the Ion Torrent S5xl platform (Thermo Fisher Scientific Inc., Waltham, MA, USA). The extracted DNA was used as the template for amplifying the V3–V4 region of the bacterial 16S rRNA gene using the following universal primers, adapter sequences, and index sequences: 341F (5′-CCT ACG GGN GGC WGC AG-3′), sample-specific 6–8-bp tag sequence, and 805R (5′-GAC TAC HVG GGT ATC TAA TCC-3′). PCR was performed using a Platinum PCR SuperMix high fidelity system (Thermo Fisher Scientific Inc.) in a final reaction volume of 27 μL, using 2.5 ng of template DNA and a 50 nM solution of each primer. The cycling conditions were as follows: initial denaturation at 94 °C for 3 min, followed by 30 cycles of denaturation at 94 °C for 30 s, annealing at 50 °C for 30 s, and elongation at 72 °C for 30 s. The amplicon libraries were further purified to remove residual primer dimers and contaminants, using the Agencourt AMPure XP DNA Purification Kit (Beckman Coulter, Brea, CA, USA), according to the manufacturer’s instructions. Samples were eluted in 15 μL of a low-EDTA Tris EDTA buffer. DNA concentration and quality and amplicon library concentration were assessed using the dsDNA High Sensitivity Assay Kit on a Qubit 4 Fluorometer instrument (Thermo Fisher Scientific Inc.). The fragment size and quality of the pooled DNA were assessed using an Agilent 2100 Bioanalyzer system (Agilent Technologies, Palo Alto, CA, USA). The concentrated particles were loaded into an Ion 530 Chip Kit (Thermo Fisher Scientific Inc.), and sequencing was performed using an Ion GeneStudio S5 system (Thermo Fisher Scientific Inc.) according to the manufacturer’s instructions [28,29]. High-throughput 16S rRNA gene amplicon sequencing analysis was performed using S5, a next-generation sequencer on the Ion Torrent S5xl platform.

#### 2.7.3. Analysis of Ion Torrent Sequencing Results

The FASTQ file containing raw 16S rRNA sequence data was obtained using Torrent Suite Software version 5.14.1.1. (Thermo Fisher Scientific Inc.). The 16S rRNA workflow module in the EzBioCloud software (ChunLab, Seoul, Korea) was used to classify individual reads by the combined use of the basic local alignment search tool and the curated Greengenes Database, which contains a high-quality library of full-length 16S rRNA sequences [30]. Reads shorter than 500 bp or those that were inappropriately paired were excluded from the analysis. Additionally, chimeras were removed from the sequence data. The sequences were clustered into operational taxonomic units at 97% identity using the QIIME pick_open_reference_otus.py, the Greengenes 13.5 reference database, and the UCLUST algorithm.

### 2.8. Statistical Analysis

The data were statistically analyzed using the SPSS software (version 20.0; SPSS, Chicago, IL, USA). Significance was determined when the *p*-value was less than 0.05. Bar graphs and plot graphs were generated using GraphPad Prism version 8.3.1 for Windows (GraphPad Software, San Diego, CA, USA). A parametric paired *t*-test was performed to compare data at two different time points and to determine the mean and standard deviations of the dependent variables of the two groups. The Wilcoxon rank-sum test was used as a non-parametric test. Spearman’s correlation coefficient was used for correlation analysis of the heatmap, and the symbol was denoted as “*ρ*”.

## 3. Results

### 3.1. General Characteristics of Participants

One patient withdrew consent at the time of enrollment and two patients withdrew consent at the time of randomization; thus, a total of three participants were excluded before the start of the study, leading to a total of 51 participants in the study. There were no dropouts until the end of the study. The participants were postmenopausal women aged more than 65 years, with an average age of 70.18 ± 2.43 years (Table 1). The vital signs and physical measurements were within normal ranges, and the participants were all healthy elderly women. All participants were non-smokers.

### 3.2. Nutrient Intake Assessment

We assessed the nutrient intake at each visit using the 24-h recall method (Appendix A). When the nutrient intakes of all groups at baseline were compared, the protein intake was found to be significantly lower in Group C (*p* = 0.025). However, there was no significant difference among the nutrient intakes in the different groups.

We compared the nutrient intake at each time point (Appendix A). In particular, when all groups were compared at the time of ingestion at 4 weeks, Group A showed a significantly high intake at 36.10 g (*p* = 0.025), followed by Group B at 27.97 g, and Group C at 25.44 g. Following 1 week of MPRO3 consumption, the nutrition intake for Group A increased from 28.54 to 32.69 g, and that of Group B increased from 28.20 to 29.90 g. In Group A, dietary fiber intake steadily increased from 28.54 g at baseline to 36.10 g following 4 weeks of MPRO3 consumption (*p* = 0.025).

### 3.3. Hematological and Chemical Blood Tests

We evaluated the safety parameters of the participants by hematological examination (Appendix A). The test comprised a red blood cell panel, a white blood cell panel, and a platelet panel. None of the parameters showed changes based on MPRO3 or placebo intake.

We evaluated the clinical effects of MPRO3 through biochemical tests (Appendix A) (lipid, renal, liver, and inflammation panels). The glucose levels did not reduce significantly when MPRO3 was administered for 1 week, but the levels decreased significantly after MPRO3 administration for 4 weeks (*p* = 0.038). Among the glucose level-related data measured at 4 weeks, the data for two participants from Group A were excluded because the participants were not fasting during measurement. The hemoglobin A1c (HbA1c), albumin (ALB), and total protein levels (TP) were maintained during the study period. The blood urea nitrogen level was significantly decreased when MPRO3 was administered for 1 week (*p* = 0.000, 0.000). Triglyceride levels decreased from 152 mg/dL at 0 weeks to 131.76 mg/dL at 4 weeks; however, it was not significant. One patient was excluded from Group B due to insufficient blood volume for zonulin measurement.

### 3.4. Assessment Using the GSRS and BSFS

We administered a questionnaire based on the GSRS and BSFS at each visit (Figure 2). The GSRS is a disease-specific instrument. The abdominal syndrome cluster included abdominal pain, a sensation of stomach emptiness, nausea, and vomiting. In the abdominal syndrome cluster, there were no changes that indicated a trend based on MPRO3 consumption or the period of intake. The reflux syndrome cluster included heartburn and reflux syndromes. In particular, the average reflux syndrome cluster score showed a decrease at 0 (1.39) and 1 week (1.17) in Groups A and B when MPRO3 was administered for 1 week (*p* = 0.012). When MPRO3 was administered for 3 additional weeks (Group A), there was a consistent reduction in the scores at 4 weeks (1.03) (*p* = 0.038). An increase was observed at 4 weeks (1.28) in Group B participants, who received the placebo from week 2 to week 4 (*p* = 0.014). The indigestion syndrome cluster included borborygmus, abdominal distension, eructation, and increased flatus. The average scores for the indigestion syndrome cluster showed a reduction at 0 weeks (1.50) and 1 week (1.26) in Groups A and B (*p* = 0.003). With MPRO3 administration for 3 more weeks (Group A), the score further decreased, as measured at 4 weeks (1.08) (*p* = 0.012). Meanwhile, in Group B, which received the placebo for 3 weeks, the score had increased at 4 weeks (1.27). The constipation syndrome cluster included the decreased passage of stool, hard stool, and a feeling of incomplete evacuation. In particular, the scores in the constipation syndrome cluster were consistently maintained at all time points (1.11, 1.09, and 1.09) in Group A participants, who were administered MPRO3 for 4 weeks. The diarrhea syndrome cluster included the increased passage of stools, loose stools, and an urgent need for defecation. In particular, in Groups A and B, the average score in the diarrhea syndrome cluster reduced at 0 weeks (1.53) and 1 week (1.25) (*p* = 0.025). At 4 weeks, after MPRO3 administration for 3 weeks, the score in Group A showed a further reduction (1.09) (*p* = 0.046). Meanwhile, Group B, who received the placebo for 3 weeks, showed a greater score at 4 weeks (1.55) (*p* = 0.043).

The BSFS comprises five items: shape, frequency, smell, size, and color (Figure 2B). In Group A, with respect to shape, the score decreased significantly at 1 week (4.47, *p* = 0.021) from that at 0 weeks (4.87) (i.e., after MPRO3 ingestion for 1 week); the score decreased significantly after 3 weeks of additional intake, i.e., at 4 weeks (4.00, *p* = 0.019). The frequency increased from that at 0 weeks (6.78) to that at 1 week (6.88), but only in Group A, and continued to increase until a higher score was obtained at 4 weeks (7.12).

### 3.5. Changes in the Taxonomic Composition, Based on Intake for 1 Week

We observed the abundance of gut microbiota before and 1 week after the administration of MPRO3 or the placebo (Figure 3). Among the typically observed bacteria, Bacteroidetes and Firmicutes became less abundant at 1 week compared to the levels at 0 weeks in Groups A, B, and Group C (Figure 3A). Conversely, the abundance of *Bifidobacterium* increased significantly (*p* = 0.002) at 1 week compared to that at 0 weeks in Groups A and B. In Group C participants who received the placebo, the abundance of *Clostridium* increased significantly (*p* = 0.002) at 1 week compared to that at 0 weeks. Beta-diversity analysis was performed to observe the similarity between samples from different groups based on the intake for 1 week (Figure 3B). The abundances in Groups A and B changed from 0 weeks to 1 week. In Group C, there was no clear distinction between the abundances at 0 weeks and 1 week.

### 3.6. Changes in the Taxonomic Composition, Based on Intake for 4 Weeks

We observed the abundance of gut microbiota before and 4 weeks after the administration of MPRO3 or the placebo (Figure 4). The abundance of *B. animalis* increased significantly (*p* = 5.25 × 10^5^) at 1 week compared to that at 0 weeks in Group A (Figure 4A). As a result, at the 1-week mark, the difference between the *B. animalis* abundances in Groups A and C, who received MPRO3, and the placebo for 4 weeks, respectively, was significant (*p* = 5.9 × 10^5^). The abundances of *B. pseudolongum*, *L. plantarum,* and *L. paracasei* were significantly higher (*p* = 0.00007 for *B. pseudolongum*; *p* = 0.031 for *L. paracasei*) at 1 week than at 0 weeks in Group A, which received MPRO3 for 4 weeks. As a result, at 1 week, the difference between the *L. paracasei* abundances in Groups A and C was significant (*p* = 0.004). Via principal coordinate analysis (PCA), the parameters in Group A at 0 weeks and 1 week were clearly distinguished (Figure 4B). Meanwhile, in Group C, there was no clear distinction between the parameters at 0 weeks and 1 week.

### 3.7. Comparison of the Results of MPRO3 Intake for 1 Week and 4 Weeks

We compared whether there was any difference in the degree of bacterial abundance when MPRO3 was ingested for 4 weeks and when it was taken for 1 week (Figure 5). The *B. animalis* abundance increased significantly (*p* = 4.3 × 10^−6^) at 4 weeks from that at 0 weeks in Group A. As a result, the *B. animalis* abundances at 4 weeks in Group A and 1 week in Group B (fed MPRO3 for 4 weeks and 1 week, respectively) showed a significant difference (*p* = 5.2 × 10^−5^) (Figure 5A). In Group A, the abundance of *B. pseudolongum* increased significantly (*p* = 0.004) at 4 weeks, compared to that at 0 weeks. *B. pseudolongum* was not observed at 1 week in Group B. The abundance of *L. paracasei* increased significantly (*p* = 0.007) at 4 weeks, compared to that at 0 weeks in Group A. At 1 week in Group B, *L. paracasei* was observed in very low abundance, and there was a significant difference between the abundance at 4 weeks in Group A and 1 week in Group B (*p* = 0.022). In Group A, the abundance of *L. plantarum* increased significantly (*p* = 0.018) at 4 weeks compared to that at 0 weeks.

In the same population, the changes in the bacteria after 1 week of intake of MPRO3 were observed after taking the placebo for 3 weeks. In addition, when MPRO3 intake was stopped, whether the intake returned to the same as before intake was analyzed (Figure 5C,D). The abundance of *Bifidobacterium* increased significantly after 1 week of MPRO3 administration (*p* = 0.016). Meanwhile, the abundance decreased at 4 weeks, after MPRO3 was discontinued and the placebo was administered for 3 weeks. The abundance of *B. longum* increased significantly after MPRO3 administration for 1 week (*p* = 0.033) but had decreased by 4 weeks.

In the PCA, Groups A and B at 0 weeks (before MPRO3 administration for 1 week) and Group B at 1 week (after MPRO3 administration for 1 week) were clearly distinguished (Figure 5B). Groups A and B at 0 weeks and Group A at 4 weeks were also clearly distinguished. In addition, for Group B, the results at 0 weeks (before MPRO3 ingestion), 1 week (after 1 week of MPRO3 ingestion), and 4 weeks (after 3 weeks of placebo ingestion) were compared. The characteristics of Group B at 0 weeks were more similar to the characteristics at 4 weeks, and not to those at 1 week, in the different groups.

### 3.8. Functional Biomarker Identification According to MPRO3 Intake

We performed an analysis to estimate the abundance of genes, using the results of the 16S taxonomic profiling and gene information for each species from the genome database (Figure 6).

On the basis of the information from the Kyoto Encyclopedia of Genes and Genomes (KEGG) database, the modules were analyzed using PICRUSt, and the pathways were analyzed using MinPath (Figure 6A). Analysis of significant modules and pathways involved in both weeks 1 and 4 of intake showed significant differences between these two groups. The module represents a function-based gene set unit that categorizes similar functions. Ten modules (cobalamin biosynthesis (*p* = 0.0091), PTS system, fructose-specific II component (*p* = 0.0407), semi-phosphorylative Entner–Doudoroff pathway (*p* = 0.0021), inositol transport system (*p* = 0.0083), inositol phosphate metabolism (*p* = 0.0294), glycerol transport system (*p* = 0.0056), pentose phosphate pathway (*p* = 0.0376), oligosaccharide transport system (*p* = 0.0294), N-glycan biosynthesis (*p* = 0.0017), and ESCRT-I complex (*p* = 0.0068)) were found to be associated with both groups, but were more significantly associated with Group A.

Following this, the metabolic pathways, in which the modules gather and function as a single unit, were analyzed. Eight pathways (fructose and mannose metabolism (*p* = 0.0407), insulin signaling pathway (*p* = 0.0091), mineral absorption (*p* = 0.0050), cortisol synthesis and secretion (*p* = 0.0376), Toll-like receptor (TLR) signaling pathway (*p* = 0.0056), sphingolipid signaling pathway (*p* = 0.0001), linoleic acid metabolism (*p* = 0.0160), and phenylalanine, tyrosine, and tryptophan biosynthesis (*p* = 0.0282)) were found to be associated with both groups, but were more significantly associated with Group A. In particular, the glycerolipid metabolism pathway (*p* = 0.0347) was not found to be associated with Group B. Conversely, the insulin resistance pathway (*p* = 0.0133) and FoxO signaling pathway (*p* = 0.0110) were found to be more significantly associated with Group B than with Group A.

We analyzed the correlation between the significantly abundant bacteria and significantly associated pathways at the 4-week time point in Group A (Figure 6B). The abundance of *B. animalis* was positively correlated with the mineral absorption pathway (*ρ* = 0.483, *p* = 0.050) and cortisol synthesis and secretion pathways (*ρ* = 0.475, *p* = 0.054), whereas it was negatively correlated with the zonulin level (*ρ* = −0.520, *p* = 0.033). The abundance of *B. pseudolongum* was positively correlated with the mineral absorption pathway (*ρ* = 0.605, *p* = 0.010) and cortisol synthesis and secretion pathways (*ρ* = 0.614, *p* = 0.009), whereas it was negatively correlated with the zonulin level (*ρ* = −0.488, *p* = 0.047). The zonulin levels were measured at 1 week in Group B and at 4 weeks in Group A; one patient was excluded from Group B, owing to an insufficient volume of serum sample. We further analyzed the correlation between *B. animalis* and *B. pseudolongum* abundances and observed a positive correlation (*ρ* = 0.497, *p* = 0.042). The correlation between the significantly abundant bacteria and associated pathways at the 1-week time point in Group B was analyzed (Figure 6B), and no significant correlation was observed.

## 4. Discussion

Intestinal microflora are involved in various metabolic processes, such as digestion, defecation, nutrient absorption, and lipid metabolism [5]. The development of various metabolic diseases, including diabetes and GI diseases, is common among the elderly [7,8]. Supplements, such as synbiotics, are a major means of improving the intestinal microflora composition to maintain intestinal homeostasis and promote health in the elderly [29]. However, further research is needed to confirm the effect of specific synbiotics, via investigation of the causal relationship between intake and beneficial effects observed in the study cohort. This study aimed to evaluate the changes in blood parameters, defecation activity, and intestinal microflora in elderly women who received synbiotics. In addition, this study aimed to observe and compare the physiological effects of synbiotics in terms of short-term and long-term intake and non-intake groups.

We observed a decrease in glucose levels in Group A after taking MPRO3 for 4 weeks. In GSRS, it was observed that reflux syndrome, indigestion syndrome, and diarrhea syndrome gradually improved as MPRO3 intake continued in Group A, and constipation was maintained. The stool shape of BSFS was observed to be improved in Group A. In Group AB, who received MPRO3 for 1 week, the overall intestinal microbiome was changed after ingestion; the abundance of Bifidobacterium increased, and the abundance of *Clostridium* decreased. In Group A, who took MPRO3 for 4 weeks, the overall intestinal microbiome changed after ingestion, and the abundance of *B. anlimalis*, *B. pseudolongum*, *L. plantarum*, and *L. paracasei* increased. When the intake of MPRO3 for 4 weeks and the intake for 1 week were compared, the abundance of *B. anlimalis*, *B. pseudolongum*, *L. plantarum*, and *L. paracasei* was relatively high in Group A after 4 weeks of intake. Bifidobacterium and *B. longum* increased after ingestion in Group B for one week and decreased again when intake was stopped and returned. When the modules and pathways involved in both groups were analyzed, all 10 analyzed modules were relatively highly involved in Group A. In particular, in terms of pathways, the FoxO pathway and the insulin resistance pathway were relatively highly involved in Group B. When the specific bacteria and pathways were correlated at each time point and each group, the mineral uptake pathway and the cortisol biosynthesis and secretion pathway in Group A were correlated with *B. animalis* and *B. pseudolongum*.

We observed that the dietary fiber intake increased, and glucose levels decreased when MPRO3 was administered for 4 weeks. On the basis of GSRS, it was found that in Group A, reflux syndrome, indigestion syndrome, and diarrhea syndrome gradually improved with continuous MPRO3 intake, whereas constipation was maintained. The stool shape (according to the BSFS scale) improved in Group A. In Groups A and B, the overall intestinal microbiome composition changed after 1 week of MPRO3 ingestion, as the abundance of *Bifidobacterium* increased and that of *Clostridium* decreased. In Group A, the overall intestinal microbiome composition changed after 4 weeks of MPRO3 ingestion, and specifically, the abundances of *B. animalis, B. pseudolongum, L. plantarum,* and *L. paracasei* increased. When the effects of MPRO3 intake for 4 weeks and 1 week were compared (Group A vs. Group B, respectively), the abundances of *B. animalis*, *B. pseudolongum*, *L. plantarum*, and *L. paracasei* were found to be relatively higher in Group A. The abundance of *Bifidobacterium* and *B. longum* increased after MPRO3 ingestion for 1 week (Group B) and decreased again when the intake was discontinued. When the modules and pathways involved in both groups were analyzed, all 10 analyzed modules showed a relatively strong association with Group A. In particular, among the pathways, the FoxO and insulin resistance pathways were relatively strongly associated with Group B. When the correlation between the abundance of specific bacteria and the pathways involved was evaluated at each time point for each group, the mineral uptake pathway and the cortisol biosynthesis and secretion pathways were correlated with the *B. animalis* and *B. pseudolongum* abundances in Group A.

The study population showed mean GSRS scores in the range of 1 to 2, suggesting that a significant proportion of study participants experienced no or mild intestinal problems (Figure 2A). Therefore, no statistical significance was observed for the GSRS scores. However, in a randomized study targeting participants with intestinal symptoms, the GSRS measurement needs to be used as the basis for evaluation [31]. On average, elderly women show more severe GI symptoms than men and have higher GSRS scores [32]. A potential explanation is that hormonal changes resulting from the menopause can increase their susceptibility to mucosal damage and interfere with tissue repair in the digestive tract [33]. Improved outcomes for diarrhea, constipation, and reflux have been reported in probiotic intervention studies [34]. In this study, the intake of a high-protein diet improved reflux, constipation, and diarrhea in the GSRS, whereas it did not improve abdominal pain and indigestion [35]. This suggests that probiotic interventions may have different effects on the GSRS parameters.

The BSFS is a key tool used for explaining bowel habits [36]. Stool consistency generally refers to the fluidity or viscosity of feces and is largely determined by the fecal water content [37,38]. Rapid intestinal transit limits the absorption of water in the stomach, leading to loose or liquid stools, whereas a slow intestinal transit results in extensive water absorption, resulting in hard stools [39]. Reportedly, there was no improvement in BSFS scores upon the consumption of lactic acid-containing fermented milk [10]. However, in this study, the BSFS score for shape improved with MPRO3 intake over 4 weeks (Figure 2B). When MPRO3 was ingested for 1 week, it was observed that there was no significant improvement in Group B in the short term. This predicts the potential for the improvement of diarrhea syndrome with Mpo3 intake for 4 weeks.

Unlike other Gram-positive Enterobacteriaceae, bifidobacteria metabolize fructose 6-phosphate phosphoketolase generated from the d-fructose 6-phosphate shunt to ferment hexoses released from oligosaccharides [40,41]. The oligosaccharides used in this reaction are suitable substrates for the growth of bifidobacteria [42,43,44]. Bifidobacteria alter the proportion of organic acids produced during carbohydrate fermentation, and some end products of this metabolic pathway can exert beneficial effects on human health [45,46]. *B. animalis* is considered a probiotic and is used as an active ingredient in functional dairy-based products, and it has been shown to be effective in reducing weight gain and improving diabetes in animal models [47]. It was also shown to control the frequency and shape of stool in adults [48]. This is consistent with the results of the present study (Figure 2). *B. pseudolongum* was shown to improve plasma triglyceride and reduce visceral fat in obese mice [49]. In the present study, the abundance of *B. animalis* increased with the intake of MPRO3 for 4 weeks. The plasma glucose levels decreased significantly. However, there was no improvement in the plasma TG levels (Appendix A). *Lactobacillus* is one of the most commonly used probiotics [50,51]. Reportedly, *Lactobacillus* is effective in inhibiting the growth of putrefactive bacteria and improving intestinal health, including symptoms such as diarrhea [52,53]. *L. plantarum* has been shown to produce antibacterial bacteriocins. *L. paracasei* is highly resistant to gastric and bile acids, and it reaches the intestine through live plaques [54]. *L. plantarum* and *L. paracasei* are known to produce short-chain fatty acids, including butyrate. Butyrate is a short-chain fatty acid that is not synthesized in the human body. It is produced via the fermentation of dietary fiber and has been reported to affect blood glucose levels and the risks of certain diseases [55]. In the present study, after 4 weeks of MPRO3 ingestion, the dietary fiber intake increased significantly (*p* = 0.025), and the abundances of *L. plantarum* and *L. paracasei* also increased. It also increased the abundance of other lactic acid bacteria, such as *Bifidobacterium*. The effects seemed to persist with ingestion for more than 3 days per week [56]. By providing an energy source for the gut microbiota, prebiotics regulates the composition and function of this microflora, and the fermentation of prebiotics by the gut microbiota produces short-chain fatty acids (SCFAs) [57]. SCFA is an important metabolite for maintaining intestinal homeostasis, including maintaining intestinal barrier integrity. The prebiotics contained in MPRO3 promote the production of beneficial bacteria, induce the production of short-chain fatty acids, and are expected to be attributable to the improvement of the intestinal environment.

In the present study, no significant increase in *Lactobacillus* abundance was observed after 1 week of MPRO3 intake. However, the abundance of *Clostridium* and Firmicutes decreased. The abundance of *Clostridium* did not decrease when a placebo was ingested for 1 week, but the abundance of Firmicutes also decreased with placebo intake (Figure 3A). The abundances of *Lactobacillus* and *Bifidobacterium* increased with MPRO3 ingestion for approximately 4 weeks, compared to those observed with MPRO3 ingestion for 1 week (Figure 4A). When MPRO3 was ingested for 4 weeks, the dietary fiber intake increased; therefore, the activity of prebiotics present in MPRO3 could be predicted. We also observed that these bacteria were enriched when MPRO3 was continuously ingested for more than 4 weeks.

A module is a unit of a gene set that categorizes similar functions, based on PICRUSt. We analyzed the modules associated with MPRO3 ingestion for 1 and 4 weeks. As MPRO3 ingestion was a common factor, only the significant modules were filtered when both groups were involved. Seven modules were filtered, and all modules had more gene sets in the group fed MPRO3 for 4 weeks (Figure 6A). Cobalamin, a component in cobalamin biosynthesis (the module with the highest abundance in this study), is also known as vitamin B12. It is produced during the fermentation of ingested animal food products by intestinal microbes, and its deficiency occurs owing to intestinal malfunction [58]. The PTS system, the fructose-specific II component module, catalyzes the phosphorylation of carbohydrates and is involved in fructose transport [59]. The semi-phosphorylative Entner–Doudoroff pathway module diversifies and activates key carbohydrate metabolic pathways [60]. Inositol in the inositol transport system and the inositol phosphate metabolism module improve insulin sensitivity, and defects in the inositol transport system increase insulin resistance [61]. In the glycerol module, glycerol promotes intestinal water absorption; therefore, it is used during acute GI diseases and constipation [62]. The pentose phosphate pathway module is associated with the supply of NADPH for the regulation of cancer-cell growth through intracellular ROS detoxification, reductive biosynthesis, and ribose biosynthesis [63].

A “pathway” refers to a metabolic pathway in which the MinPath modules work as a single function. We observed the pathways and zonulin measurement results with MPRO3 ingestion for 1 and 4 weeks. Nine pathways were filtered; two were more abundant in the 1-week intake group, and seven were more abundant in the 4-week intake group (Figure 6A). The fructose and mannose metabolism pathways enhance immunoreactive insulin secretion, glucose disposal, and gastric inhibitory polypeptide secretion [63]. In the present study, the insulin signaling pathway was more strongly associated with the 4-week intake group than the 1-week intake group (Figure 6A). The insulin resistance pathway was more strongly associated with the 1-week intake group (Figure 6A). This result is consistent with the results of previous studies on animal models [64]. The FoxO signaling pathway mediates the inhibitory actions of aging, oxidative stress, and insulin or insulin-like growth factor [65,66]. FoxO signaling was more strongly associated with the 1-week intake group, and its association with the 4-week intake group was weaker (Figure 6A). Regarding the association with the glycerol transport system module, along with the glycerolipid metabolism pathway, it showed no association with the 1-week intake group and was only associated with the 4-week intake group. In terms of the TLR signaling pathway, TLR maintains homeostasis among commensal microbiota, explaining the intestinal regulatory system [67]. Reportedly, the expression of zonulin, an indicator of intestinal epithelial wall function, is improved upon TLR modulation [67,68]. In the present study, the TLR signaling pathway was more involved after 4 weeks of ingestion; additionally, the zonulin levels were lower. Regarding the sphingolipid signaling pathway, sphingolipid signaling affects insulin sensitivity, lipid metabolism, and inflammatory processes [69].

We analyzed the correlation between the abundance of specific bacteria and the activation of different pathways after 1 and 4 weeks of MPRO3 ingestion (Figure 6B). After 1 week of ingestion, no significant correlation was observed between the abundance of specific bacteria and the pathways. After 4 weeks of ingestion, we observed a correlation between the abundances of *B. animalis* and *B. pseudolongum* and mineral absorption, cortisol synthesis and secretion, and zonulin. Vitamins and minerals from digested food are extracted in the small intestine, and most of these are also absorbed in the small intestine [70,71]. After 4 weeks of MPRO3 ingestion, the abundance of *B. animalis* and the activation of the mineral absorption pathway increased, and a positive correlation was observed between the two. In particular, increased water absorption in the small intestine ameliorates diarrhea; in our study, appearance values in the BSFS and diarrhea values in the GSRS showed improvement. Cortisol (in the cortisol synthesis and secretion pathway) is an indicator of stress [72]. Recently, there have been studies on the correlation between cortisol and probiotics but no significant results have been reported [73]. Leakage in the intestine is caused by the dysregulation of zonulin and impaired intestinal permeability, which contributes to the pathogenesis of GI disorders [74]. There is a difference of opinion on the correlation between the development of a leaky gut and probiotic intake [75]. In the present study, the zonulin levels were lower after 4 weeks of MPRO3 intake than after 1 week of MPRO3 intake. However, it is difficult to confirm a causal relationship because zonulin expression was not measured or analyzed during all visits. Therefore, based on the findings of our study, we suggest the possibility of a reduction in zonulin production in response to prolonged MPRO3 ingestion.

In the present study, we observed changes in blood parameters, bowel activity, and intestinal microflora, based on MPRO3 intake, and compared its short- and long-term effects. This study had certain limitations, and the use of antibiotics and probiotics (except MPRO3) was not allowed. First, the genes in the observed modules and pathways indicated abundance and not expression. This was an analysis to estimate the abundance of genes using the results of 16S taxonomic profiling and gene information for each species in the genome database. However, since insulin, mineral metabolism, and zonulin observed in each module and pathway indicated improvement, we can predict the possibility of further improvement with MPRO3 ingestion for more than 4 weeks. Second, all participants were female. The gut microbiome can be affected by multiple variables, and gender and inter-individual variability may vary, which makes it challenging to match groups with equal species-level proportions in clinical trials. We focused on the outcomes in older women by selecting only women of a certain age group to narrow down the range of variables, including hormones. Third, all participants were residents of a specific area. Although adherence to and compliance with the study guidelines among the participants was very high, a follow-up study is needed to generalize the results of this paper.

## 5. Conclusions

In this study, we evaluated the changes in blood parameters, defecation activity, and intestinal microflora composition in response to MPRO3 ingestion and compared the effects of short-term and long-term MPRO3 intake. Changes in the intestinal microbiome were induced, even with MPRO3 ingestion for only 1 week. However, after 1 week of ingestion, barring some changes in the intestinal microflora, improvement in the other variables was not observed, and, when intake was discontinued, the variables returned to their original state. Four weeks of MPRO3 intake reduced the glucose levels and reflux (GSRS), improved digestion and diarrhea, and also improved stool appearance (BSFS). *B. animalis, B. pseudolongum*, *L. paracasei*, and *L. plantarum* were found to be relatively abundant. In addition, the abundances of *B. animalis* and *B. pseudolongum* showed a positive correlation with the mineral absorption and cortisol metabolism pathways and a negative correlation with zonulin levels. Therefore, we suggest that MPRO3 may improve bowel activity and movement, reduce fasting blood glucose levels, and activate metabolism when ingested for more than 4 weeks.

## Figures and Tables

**Figure 1 microorganisms-10-00088-f001:**
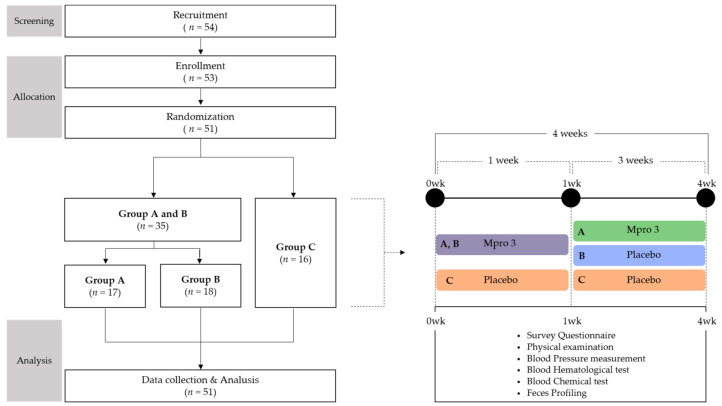
Study Design.

**Figure 2 microorganisms-10-00088-f002:**
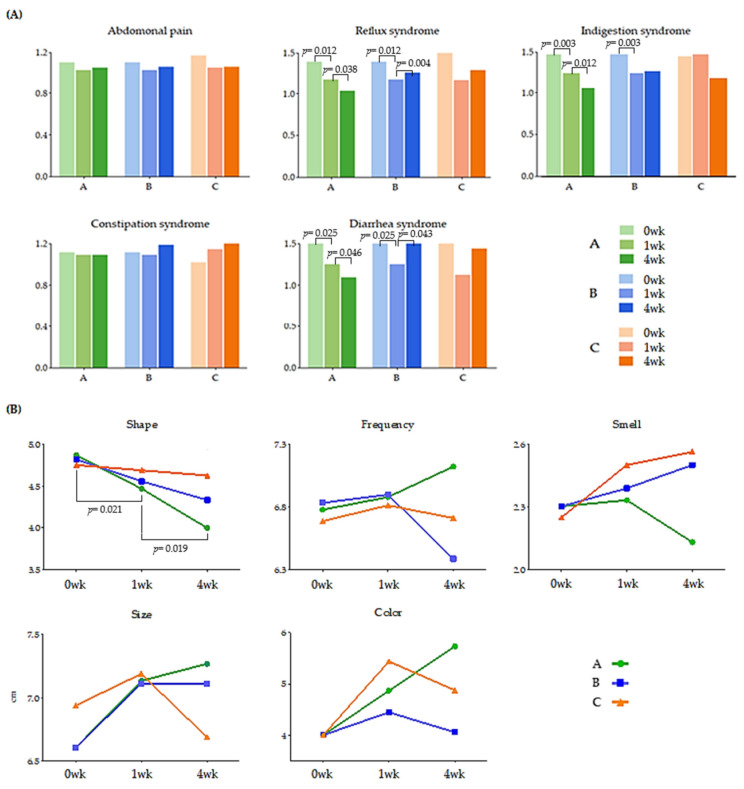
Change of GSRS (**A**) and BSFS (**B**). Gastrointestinal symptoms rating scale (GSRS) and Bristol stool form scale (BSFS). The values of all groups: TP 1 (at 0 weeks), TP 2 (at 1 week), and TP 3 (at 4 weeks) are shown. Using the Wilcoxon rank-sum test, each group was compared, and the significance (*p*-value) was verified.

**Figure 3 microorganisms-10-00088-f003:**
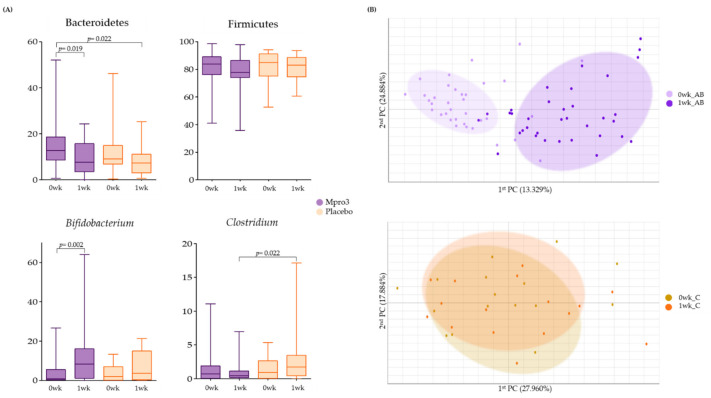
Bacterial changes in Groups A and B when taking MPRO3 for 1 week and the placebo for 1 week. Using the Wilcoxon rank-sum test, each group was compared, and significance (*p*-value) was verified. (**A**) Comparison of the abundance of specific bacteria, based on read count. (**B**) Principal coordinate analysis (PCA), showing a similarity test, based on the bacterial abundance of all samples within each group.

**Figure 4 microorganisms-10-00088-f004:**
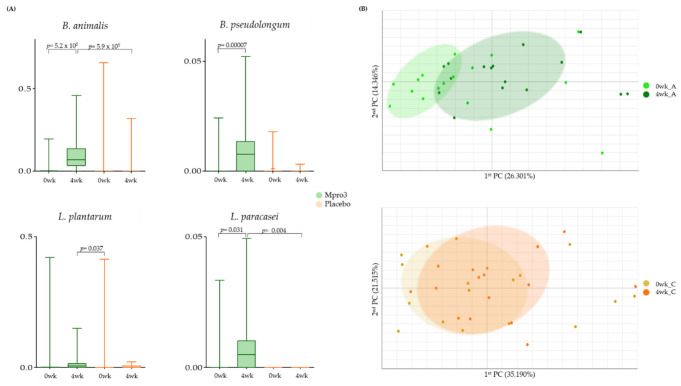
Bacterial changes in Group A taking MPRO3 for 4 weeks and the placebo for 4 weeks. Using the Wilcoxon rank-sum test, each group was compared, and significance (*p*-value) was verified. (**A**) Comparison of the abundance of specific bacteria, based on read count. (**B**) Principal coordinate analysis (PCA), showing a similarity test, based on the bacterial abundance of all samples within each group.

**Figure 5 microorganisms-10-00088-f005:**
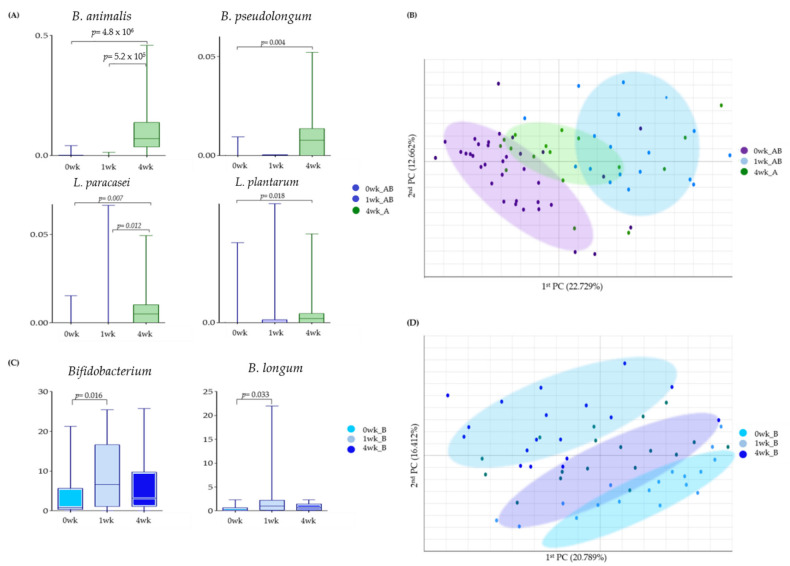
Bacterial changes in Groups A and B taking MPRO3 for 1 week and Group A taking MPRO3 for 4 weeks. Using the Wilcoxon rank-sum test, each group was compared, and significance (*p*-value) was verified. (**A**) Time-point comparison of participants taking MPRO3. Comparison of the abundance of specific bacteria, based on read count. (**B**) Time-point comparison of participants taking MPRO3. Principal coordinate analysis (PCA), showing a similarity test based on the bacterial abundance of all samples within each group. (**C**) Time-point comparison of Group B. Comparison of the abundance of specific bacteria, based on read count. (**D**) Time-point comparison of Group B. Principal coordinate analysis (PCA), showing a similarity test based on the bacterial abundance of all samples within each group.

**Figure 6 microorganisms-10-00088-f006:**
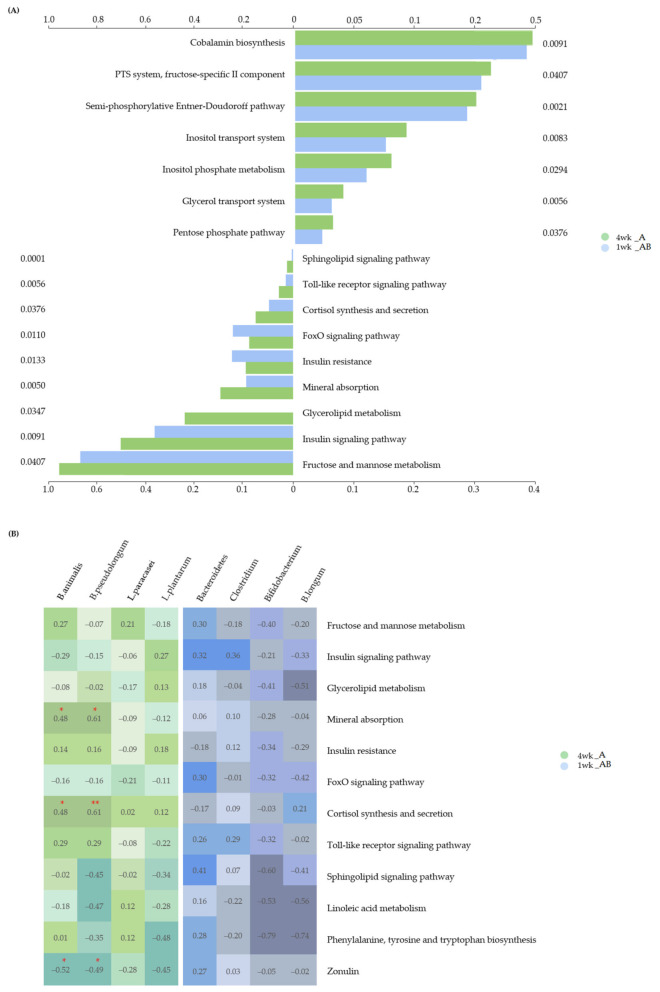
Difference between 1 week and 4 weeks of intake of MPRO3 in Module and Pathway. (**A**) Based on the KEGG database, the module was predicted using PICRUSTs, and the pathway was predicted using MinPath. (**B**) Spearman correlation analysis between predicted pathways and significant specific bacteria by intake period. * *p* > 0.05. ** *p* > 0.01.

**Table 1 microorganisms-10-00088-t001:** Basic Information in wk 0.

		Mean		Standard Deviation	
*n* =		51		0	
Age	(years)	70	0.18	4	0.40
Sex	(Female, *n* =)	51		0	
Post-menopause	(*n* =)	51		0	
Pulse rate	(min)	80	0.39	10	0.18
Systolic Blood Pressure	(mmHg)	133	0.43	15	0.83
Diastolic Blood Pressure	(mmHg)	69	0.88	10	0.83
Height	(cm)	155	0.36	5	0.49
Weight	(kg)	57	0.24	6	0.82
Body Mass Index	(kg/m^2^)	23	0.76	3	0.04
Smoking		0			
Current		0			
Past		0			

## Data Availability

The data presented in this study are available in the article and Appendix A. The raw data are available upon a reasonable request from the corresponding author.

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
