# Peer review of "Intake of MPRO3 over 4 Weeks Reduces Glucose Levels and Improves Gastrointestinal Health and Metabolism"

_microorganisms, 2021, doi:10.3390/microorganisms10010088_

Round 1

Reviewer 1 Report

This manuscript is interesting and provides new data concerning the long- and short-term effects of  Mpro3 ingestion on changes in blood parameters, defecation activity, and intestinal microflora composition in the population of elderly women. This article gives a balanced view of the topic and the overall quality of this manuscript is high based on a recent, original publication thus presenting a truly unique viewpoint. It can be stated that the obtained by the authors results  add to existing knowledge, and undoubtedly will be of interest to a journal's readers. The manuscript’s figures and table are very informative and possess clear and concise legend caption. This article is generally well written and clearly presented.

Author Response

We appreciate your careful review of our manuscript and insightful remarks. 
We will continue to study hard for this field.
Thank you.

Reviewer 2 Report

The manuscript by Lee et al. describes an intervention study testing the effects of a synbiotic drink Mpro3 on metabolism and digestive health in elderly women.  In general, the study was well-received and the manuscript was nicely written.  A few points shown below aim to improve clarity of the manuscript.

  1. It would be nice to introduce Mpro3 in the Introduction rather than waiting till the Methods.
  2. It is unclear from the manuscript why the groups were set up that way. Specifically, please provide the rationale for group B.
  3. The Mpro3 intervention groups had fibers but the placebo did not. Would this have impacted the results? Suggest more accurate description (Mpro3 and fibers rather than Mpro3 alone) and additional discussion on the potential effects of fibers.
  4. Fig. 2 is important because it shows the benefits of Mpro3 on digestive health. However, the statistical analysis was inadequate, or at least poorly described.  Were any differences in GSRS significant?  In addition, some BSFS scores may be different among groups at specific time points but not labeled or discussed.  
  5. Fig. 5: It is unclear why these groups were chosen for comparisons. Also, would 1wk_B be the same as 1wk_AB?  Suggest presenting data on all possible comparisons rather than only selecting some groups/time points.
  6. Fig. 6: Again, it is unclear which comparisons were made. For 1wk and 4wk, were the placebo and Mpro3 groups combined? Or, was it the comparison between 1wk_AB and 4wk_A?  Suggest comparing groups A, B and C at 4wk that would be more interesting at least to this reader.

Author Response

Thank you for your careful review of our manuscript and insightful comments.
